# Lessons from the Pandemic: Role of Percutaneous ECMO and Balloon Atrial Septostomy in Multi-System Inflammatory Syndrome in Children

**DOI:** 10.3390/jcm13082168

**Published:** 2024-04-09

**Authors:** Ranjit R. Philip, Claire Sentilles, Jason N. Johnson, Anthony Merlocco, Karthik Ramakrishnan, Kaitlin A. Ryan, Umar Boston, Shyam Sathanandam

**Affiliations:** 1Division of Pediatric Cardiology, Department of Pediatrics, University of Tennessee Health Science Center, 50 N Dunlap St., Memphis, TN 38103, USA; jjohn315@uthsc.edu (J.N.J.); amerlocc@uthsc.edu (A.M.); kbalduf1@uthsc.edu (K.A.R.); ssathan2@uthsc.edu (S.S.); 2The Heart Institute, Le Bonheur Children’s Hospital, 51 N Dunlap St., Memphis, TN 38103, USA; kramakr2@uthsc.edu (K.R.); uboston@uthsc.edu (U.B.); 3Division of General Pediatrics, Department of Pediatrics, University of Tennessee Health Science Center, 50 N Dunlap St., Memphis, TN 38103, USA; csentill@uthsc.edu; 4Division of Pediatric Cardiothoracic Surgery, Department of Surgery, University of Tennessee Health Science Center, 901 Madison Ave. Memphis, TN 38163, USA

**Keywords:** ECMO, MIS-C, COVID-19

## Abstract

Multi-system inflammatory syndrome in children (MIS-C) in the setting of COVID-19 can be associated with severe cardiopulmonary dysfunction. This clinical deterioration may sometimes necessitate veno-arterial extracorporeal membrane oxygenation (VA-ECMO) support. We describe an algorithmic approach including the role of balloon atrial septostomy in this cohort. This is the first reported series of percutaneous VA-ECMO in pediatric patients with MIS-C for better outcomes. The lessons from this approach can be replicated in other pediatric clinical conditions and adds to the armament of multiple pediatric specialties.

## 1. Introduction

Coronavirus Disease 2019 (COVID-19) is caused by the severe acute respiratory syndrome coronavirus 2 (SARS-CoV-2). Initial reports suggested that this disease was mild in children and the majority were asymptomatic [1]. However, in April 2020, the first reports in children of a hyperinflammatory syndrome hemodynamically characterized by shock associated with being positive for COVID-19 came to light [2]. The clinical features appeared to be similar to atypical or incomplete Kawasaki disease, toxic shock syndrome, or bacterial sepsis. In addition to the hyperinflammatory state, this syndrome was characterized by injury to multiple organ systems including the heart [3]. The United States Centers for Disease Control and Prevention named this multisystem inflammatory syndrome in children (MIS-C) [4]. MIS-C secondary to COVID-19 has been shown to be associated with significant morbidity and prolonged intensive care admission [5]. From the time of its initial description until March 2022, more than 7880 reported cases of MIS-C have been described throughout US, resulting at least 66 known deaths [6]. Hemodynamic and cardiopulmonary decompensation can occur rapidly in these children if not identified and managed in a timely fashion. Extra-corporeal membrane oxygenation (ECMO) is employed to support patients with severe cardiopulmonary failure to avoid irreversible organ injury. A few cases of veno-arterial (VA) ECMO using open surgical cannulation for severe MIS-C patients presenting in cardiogenic shock have been performed, [7,8,9] including at our center, with variable outcomes.

At our institution, we adopted a protocol for COVID-19 [10] and MIS-C patients [11]. We also instituted a protocol for the cohort presenting in cardiogenic shock. Time-saving measures including percutaneous VA-ECMO cannulation, followed by immediate balloon atrial septostomy (BAS), were instituted in children with MIS-C based on clinical presentation, a left ventricular ejection fraction (LVEF) <25% by echocardiography, and a vasoactive inotropic score (VIS) >25. We describe the outcomes of five pediatric patients with MIS-C complicated by severe cardiopulmonary dysfunction who required VA-ECMO support from a cohort of 162 patients admitted to the hospital between April 2020 and October 2021.

## 2. Case Presentation

All five patients presented with symptoms and physical exam findings consistent with heart failure or respiratory failure secondary to MIS-C in the setting of recent COVID-19 exposure and positive COVID-19 antibodies. The demographics, presenting symptoms, echocardiographic findings, and outcomes of these patients are presented in the Table 1. All patients had elevated inflammatory markers, liver enzymes, troponin, BNP, and D-dimer. They were treated with corticosteroids and IVIG for MIS-C. Echocardiograms prior to ECMO cannulation demonstrated severely decreased LVEF except for patients 1 and 2 who had diastolic dysfunction and respiratory failure. These five patients required mechanical ventilation secondary to LV dysfunction and pulmonary edema, refractory hypotension, and persistent metabolic acidosis requiring inotropic support.

VA-ECMO cannulation was initiated in all five cases due to severe, persistent hypotension requiring multiple inotropic support with VIS >25 and LVEF <25%, or persistent metabolic acidosis leading to other end organ dysfunction. In the first patient, VA-ECMO cannulation was attempted using standard surgical cut-down with direct cannulation of the internal jugular and carotid arteries. However, the carotid artery developed severe vasospasm during the surgical dissection. Therefore, a cut-down was performed to cannulate the femoral artery instead. The patient developed acute kidney injury (AKI) requiring continuous renal replacement therapy (CRRT). The attempted carotid cannulation led to cerebral hemorrhage and neurologic complications leading to prolonged hospitalization for 123 days with the patient eventually being transferred to a rehabilitation facility. This patient did not undergo BAS, which prolonged the days on the ventilator; ultimately, they required tracheostomy for prolonged mechanical ventilation.

The second patient also underwent open surgical VA-ECMO cannulation of the common carotid artery and the internal jugular vein. This patient underwent immediate BAS, which improved the pulmonary edema by decompressing the left atrium. However, the cannulation time was prolonged (58 min), with the patient ultimately developing a stroke involving the middle cerebral artery territory as well as thrombosis of the right common carotid artery following ECMO decannulation 5 days later. This patient also developed neurologic sequela requiring transfer to a rehabilitation facility after 28 days of hospitalization. Based on the outcomes of these two patients, an institutional protocol was established to ensure time saving measures for better patient outcomes. This led to percutaneous VA-ECMO cannulation and immediate BAS in the next three patients.

All percutaneous VA-ECMO cannulations were performed in the lower extremity vessels under ultrasound guidance with an additional limb reperfusion cannula attached to the arterial cannula (Figure 1). Figure 1 shows Patient 4 with a 25 Fr venous cannula in the right femoral vein. Insertion of a 19 Fr arterial cannula in the left femoral artery and a 6 Fr cannula for distal limb perfusion simultaneously. The time from ECMO activation to ECMO support was 13, 17 and 34 min, respectively, suggesting that percutaneous cannulation was achieved rapidly. All three patients who were cannulated percutaneously underwent BAS (Figure 2) within 2 h of ECMO support for immediate left atrial decompression. This was achieved through the percutaneous approach via transseptal puncture using real-time trans-esophageal echocardiography guidance. An atrial level communication was made using radio-frequency perforation of the atrial septum. The atrial level communication was slowly dilated using serial dilations with sequentially increasing balloon sizes. The adequacy of the atrial septum was then assessed by angiography and trans-esophageal echocardiography. There were no procedural complications, including no evidence for limb hypoperfusion. Patient 3 required CRRT for AKI, which developed even prior to cannulation. The number of days on ECMO was 3–8 days for these patients, with all of them transferred out of the ICU within 4 days after decannulation. All three patients were eventually discharged home with recovered cardiac function.

## 3. Discussion

This series is the first report of the successful use of percutaneous VA-ECMO to support children developing cardiorespiratory collapse from MIS-C in the setting of COVID-19. This report emphasizes the rare but serious consequences of severe rapid clinical deterioration associated with COVID-19 in children. In the cohort of 162 patients, 5 needed ECMO support. Of the 157 that did not need ECMO, there were patients with depressed cardiac function that improved with medical therapy such as IVIG and steroids. What distinguished these five patients from the rest of the cohort was the rapid clinical and cardiopulmonary deterioration that needed emergency support. This report highlights the life-saving role of VA-ECMO in these patients while anticipating reversal of the inflammatory response at a time when there was a shortage of resources in terms of medical personnel. Time saving measures including percutaneous VA-ECMO cannulation along with emergency BAS were the key features that improved outcomes for these children at least in our hospital. 

Open surgical VA-ECMO cannulation is the current standard in children who have very limited experience with percutaneous cannulation [12]. However, with availability of smaller percutaneous cannulas that can be advanced over a guidewire, the percutaneous approach to VA-ECMO cannulation is appealing. Percutaneous VA-ECMO could potentially offer faster cannulation and uninterrupted CPR during cannulation in addition to increasing accessibility to more providers (interventional cardiologists, radiologists, and intensivists), especially when a surgeon is not available for cannulation. Obese children are more commonly affected by MIS-C, which makes the percutaneous route with ultrasound guidance a good alternative to open cannulation from a technique perspective for these patients. 

With severe shock and cardiovascular collapse, identification of blood vessels for cannulation is challenging. The use of ultrasound for access of vessels in the cardiac catheterization laboratory has significantly improved the time taken for and the safety profile of obtaining arterial and venous access [13]. Early recognition of the potential need for ECMO helps with a more efficient cannulation process. There is a lower incidence of cannula site bleeding with the percutaneous route compared to open cannulation, possibly due to the larger incision required for open cannulation [14]. For similar reasons, there is a higher rate of vascular site repair with open surgical cannulation [15]. Complications from blood clots are also more common when ECMO is achieved through the open surgical route in comparison to the percutaneous route [16]. Patients with MIS-C are already at a higher risk of thrombotic complications especially with left ventricular dysfunction and coronary artery aneurysms [17,18,19]. Tailored anti-coagulation strategies have been suggested by groups when there is significant left ventricular dysfunction and coronary artery aneurysms. The degree of D-dimer elevation has been used by groups to determine the need for anti-coagulation [20]. This can further complicate anti-thrombotic strategies whilst on ECMO. Although cannula site bleeding and thrombotic complications are likely to be less common with percutaneous cannulation in comparison to open cannulation, concerns about limb ischemia remain [21]. Limb ischemia could be avoided by introducing a distal limb perfusion cannula at the time of the arterial cannulation. It is almost impossible to insert the perfusion cannula after femoral arterial cannulation; this must be accomplished simultaneously. We achieved this by introducing two guide wires concurrently into the femoral artery, one passing towards the heart and another towards the feet, to synchronously introduce the femoral arterial and the distal limb perfusion cannulas at the same time. With continued technical evolution and advances in technology, adoption of percutaneous VA-ECMO in pediatrics should be feasible and complications can be minimized.

Immediate BAS after VA-ECMO cannulation must be considered to decompress the left atrium to prevent or treat the flash pulmonary edema that develops with rising left ventricular end diastolic pressure [22]. This may preserve lung compliance and can potentially decrease the number of days on ECMO, thereby improving survival. One might argue that we could have attempted BAS prior to consideration of ECMO. This is also a good option when left ventricular end diastolic pressures are high. However, due to the rapid clinical deterioration in these patients, ECMO was carried out prior to BAS. 

This report reiterates the fact that ECMO should be strongly considered in a small cohort of MIS-C patients since the inflammatory process is transient, as evidenced by the low average number of days on ECMO in this series. With newer COVID-19 variants on the horizon and vaccine adoption rates in the pediatric population being low, children continue to be the most vulnerable group in society. This is also of grave importance during a pandemic that has stretched medical resources in terms of the number of available ICU beds and staffing shortages [23]. These lessons from the pandemic can be extended to sepsis and other inflammatory conditions featuring cardiopulmonary compromise. This necessitates collaboration between ECMO providers, pediatric intensivists, interventional cardiologists, and pediatric and cardio-thoracic surgeons to provide accessibility and implement an organized, tailored, institutional approach that is essential for improving outcomes. 

## 4. Conclusions

The use of percutaneous VA-ECMO in pediatric patients with MIS-C is feasible and a good alternative to open surgical ECMO. Immediate atrial septostomy for left atrial decompression should be considered for these patients. A programmatic approach to specific disease processes like MIS-C and possibly extension to sepsis and other inflammatory conditions is essential for good outcomes.

## Figures and Tables

**Figure 1 jcm-13-02168-f001:**
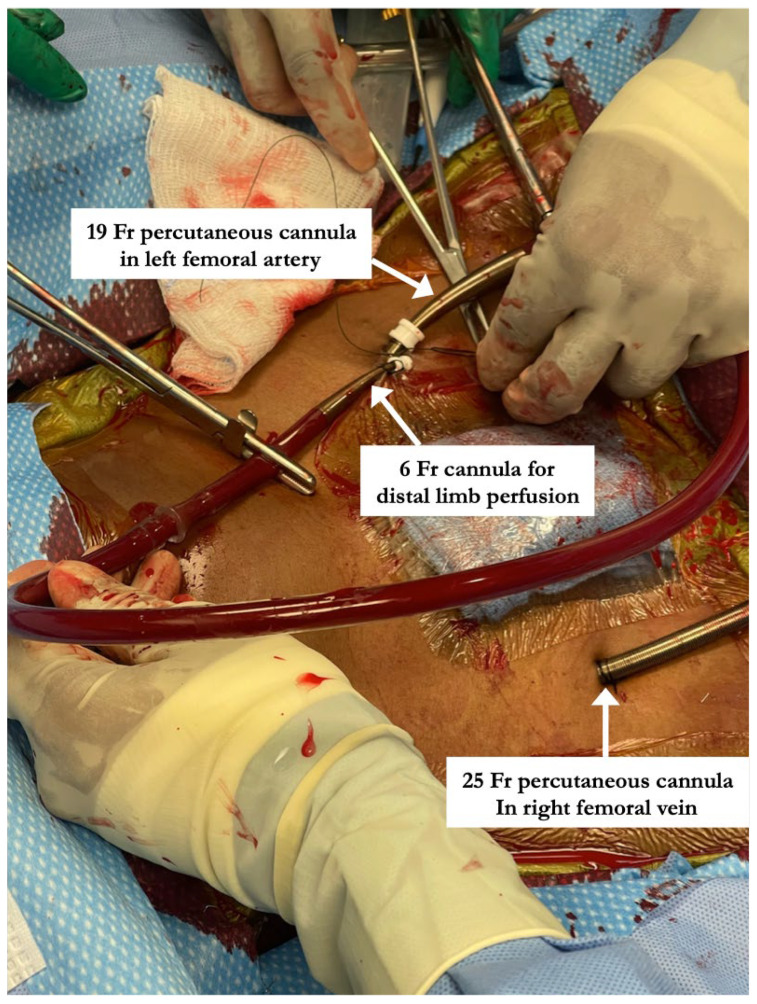
Steps involved in percutaneous arterio-venous ECMO cannulation including placement of limb perfusion cannula. “Patient 4” with a 25 Fr venous cannula in the right femoral vein. Insertion of a 19 Fr arterial cannula in the left femoral artery and a 6 Fr cannula for distal limb perfusion simultaneously.

**Figure 2 jcm-13-02168-f002:**
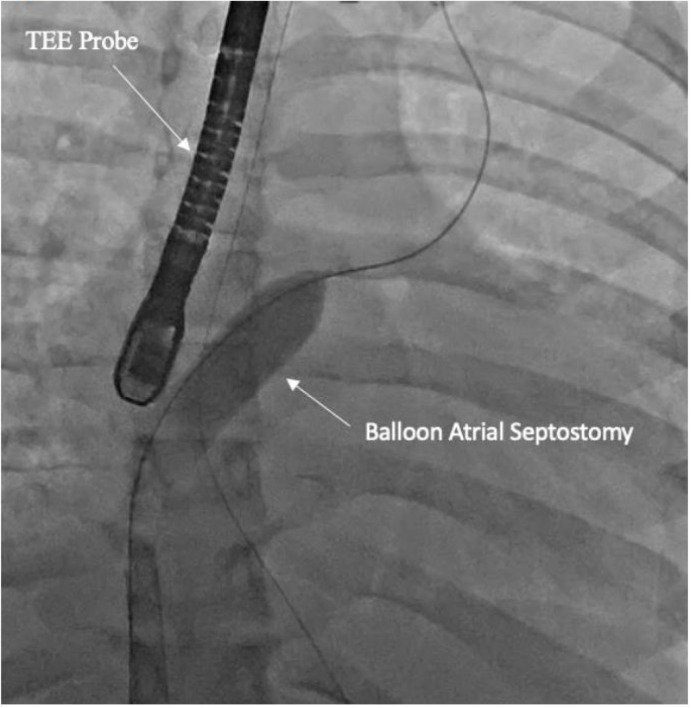
Fluoroscopy showing balloon atrial septostomy for left atrial decompression. Transesophageal (TEE) probe is seen for echo-guidance.

**Table 1 jcm-13-02168-t001:** Management and outcomes.

Patient #	Age (y)/Sex/Wt (kg)	Medical History	Initial Presentation	Echo Prior to ECMO	ECMO Type	Atrial Septostomy Performed	Length of Treatment	Outcome
1	Age: 12 Sex: M Wt: 104	Obesity, asthma	Symptoms: Fever, vomiting, diarrhea, abdominal pain, decreased appetite, headache, neck pain, altered sense of taste Echo on admission: Mild ectasia of the proximal LAD; trivial pericardial effusion; LVEF 55–60%	Hyperdynamic biventricular systolic function, decreased ventricular filling time; ectasia of the proximal LAD	Surgical VA-ECMO (right common femoral artery, right internal jugular vein) Cannula size: 25 Fr (venous), 21 Fr (arterial) VIS: 26	No	Admission to ECMO days:2 Activation to ECMO flow time: 27 min Days on ECMO: 6 Days in ICU after ECMO: 11 Days in Hospital: 123	LVEF: 55–60% Cerebral hemorrhage AKI—requiring CRRT Tracheostomy Transferred to rehabilitation facility
2	Age: 3 Sex: F Wt: 16	None	Fever, vomiting, decreased activity, rash Echo on admission: LVEF 50–55%	LVEF 50–55%	Surgical VA-ECMO (right carotid artery, right internal jugular vein) Cannula size: 19 Fr (venous), 14 Fr (arterial) VIS: 48	Yes	Admission to ECMO days:5 Activation to ECMO flow time: 58 min Days on ECMO: 5 Days in ICU after ECMO: 15 Days in hospital: 28	LVEF: 65–70% Septic shock, right MCA stroke, acquired ASD from atrial septostomy- later device closure, DVT of right iliac vein, thrombosis of right common carotid artery Transferred to rehabilitation facility
3	Age: 17 Sex: F Wt: 76	None	Fever, nausea, vomiting, diarrhea, abdominal pain, cough, congestion, headache, fatigue, myalgias, syncope, lightheadedness Echo on admission: LVEF 40–45%; nonspecific LV diastolic dysfunction	LVEF 25–30%	Percutaneous VA-ECMO (right femoral artery, left femoral vein) Cannula size: 25 Fr (venous), 21 Fr (arterial), 6 Fr (reperfusion) VIS: 23	Yes	Admission to ECMO days:3 Activation to ECMO flow time: 13 min Days on ECMO: 3 Days in ICU after ECMO: 4 Days in hospital: 12	LVEF: 55–60% Discharged home
4	Age: 17 Sex: M Wt: 75	Asthma	Fever, nausea, vomiting, diarrhea, abdominal pain, decreased appetite, headache, rash, myalgias, conjunctivitis Echo on admission: LVEF 45–50%; severe tricuspid regurgitation	LVEF 30–35%; severe tricuspid regurgitation, moderate bilateral atrial dilation, and mitral insufficiency	Percutaneous VA-ECMO (right femoral vein, left femoral artery) Cannula size: 25 Fr (venous), 19 Fr (arterial), 6 Fr (reperfusion) VIS: 28	Yes	Admission to ECMO days:4 Activation to ECMO flow time: 17 min Days on ECMO: 5 Days in ICU after ECMO: 4 Days in hospital: 23	LVEF: 60–65% DVT, AKI—requiring CRRT Discharged home
5	Age: 12 Sex: M Wt: 45	None	Fever, vomiting, abdominal pain, headache, chest pain, fatigue, palpitations Echo on admission: Severe left ventricular systolic dysfunction (LVEF 20%)	LVEF 20%	Percutaneous VA-ECMO (right femoral vein, left femoral artery) Cannula size: 21 Fr (venous), 17 Fr (arterial), 6 Fr (reperfusion) VIS: 25	Yes	Admission to ECMO days:1 Activation to ECMO flow time: 34 min Days on ECMO: 8 Days in ICU after ECMO: 4 Days in hospital: 31	LVEF: 50–55% Arterial thromboembolism, GI bleed, intra-abdominal abscess Discharged home

## Data Availability

The original contributions presented in the study are included in the article, further inquiries can be directed to the corresponding author/s.

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
