# Peer review of "Lessons from the Pandemic: Role of Percutaneous ECMO and Balloon Atrial Septostomy in Multi-System Inflammatory Syndrome in Children"

_jcm, 2024, doi:10.3390/jcm13082168_

Round 1

Reviewer 1 Report

Comments and Suggestions for Authors

A nice and well written overview of a select patient category where the concept of, in my opinion, two relative new approaches in VA ECMO are discussed. Namely 1) the percutaneous cannulation in pediatrics and 2) rapid left atrial decompression.

The select patient category is the severe hemodynamically compromised child with MIS-C. These patients were seen during the covid pandemic, but to the best of my knowledge these cases are hardly seen in current time. Therefore, might the authors not expand their concept? Is MIS-C a model for general sepsis for instance? Is the title lacking the concept of rapid atrial decompression? Abovementioned comments as a suggestion for the authors, not a critique. 

I am missing methodology on the acute balloon atrial septectomy. I am guessing that all were done by the interventional cardiologist; whereas percutaneous ECMO cannulation was done by another specialist? The video of the procedure is not available during this review. It looks like a dilatation balloon was used. There is no text describing the procedure. Did the dilated defect remain patent? How was the defect size/ballon size chosen? Need for a stent? Passage of the intra-atrial septum via transseptal puncture? Need for an atrial closure device in follow-up?

In the adult approach an intra-aortic balloon pump or a percutaneous LV assist device are used. Some of the presented cases where in their late teens. Was this approach also considered?

I congratulate the authors on their impressive clinical work and the well written paper of their excellent outcomes. 

Author Response

Thank you for the invaluable review by reviewer 1: 

The select patient category is the severe hemodynamically compromised child with MIS-C. These patients were seen during the covid pandemic, but to the best of my knowledge these cases are hardly seen in current time. Therefore, might the authors not expand their concept? Is MIS-C a model for general sepsis for instance? Is the title lacking the concept of rapid atrial decompression? Above mentioned comments as a suggestion for the authors, not a critique. 

Response: These are great points. This should be expanded to general sepsis as indicated. I have added this to the discussion.  The reason we focussed on the pandemic and MISC was because of the puacity in resources at the time and to show the tailored approach with faster access due to rapid clinical deterioration. We did not include rapid atrial decompression in the title due to the length of the title but I have added this now. 

I am missing methodology on the acute balloon atrial septectomy. I am guessing that all were done by the interventional cardiologist; whereas percutaneous ECMO cannulation was done by another specialist? The video of the procedure is not available during this review. It looks like a dilatation balloon was used. There is no text describing the procedure. Did the dilated defect remain patent? How was the defect size/ballon size chosen? Need for a stent? Passage of the intra-atrial septum via transseptal puncture? Need for an atrial closure device in follow-up?

Response: Balloon atrial septostomy was done by interventional cardiology in the cathlab. The same physician assisted in access for percutaneous cannulation with the surgeon. A dilation balloon was used which opened the septum. Stent placement was not needed unlike what is needed for a thick posteriorly deviated septum seen in hypoplastic left heart syndrome. The passage of the intra-atrial septum was with transseptal puncture. We have added these good points to the Management section. Closure device was needed only in patient 2 ( This is in the Table). 

Thank you for your suggestions.

In the adult approach an intra-aortic balloon pump or a percutaneous LV assist device are used. Some of the presented cases where in their late teens. Was this approach also considered?

This was not considered in these kids and may have been a viable option.

Reviewer 2 Report

Comments and Suggestions for Authors

I would like to congratulate authors for this novel manuscript. But I need some clarifications.

1) How were the 5 patients needing ECMO support different from rest 157 patients ? This point needs to be stressed .

2) The initial 2 patients had near normal LV ejection fraction. Why was ECMO considered for them ?

3) Was BAS done before ECMO was considered ? Were the patients reassessed after BAS for clinical improvement ? Did any patient who did not require ECMO, underwent BAS ?

4) What was the response of Steroids/ IVIG among all patients ?

5) Was Kawasaki's disease considered in any patients ?

6) Did any patient who had significant LV dysfunction at presentation  were treated without ECMO?

I think these will make your manuscript complete. 

Author Response

Thank you for these thoughtful comments. 

1) How were the 5 patients needing ECMO support different from rest 157 patients ? This point needs to be stressed .

These 5 patients had cardiopulmonary compromise with higher VIS scores with higher ventilation requirements needing ECMO. There were children with low EFs that did not need ECMO as their clinical status did not deteriorate. We have added this to the discussion. 

2) The initial 2 patients had near normal LV ejection fraction. Why was ECMO considered for them ?

As rightly pointed out, the EF in the 1st two patients were borderline. They predominantly had a SIRS response with respiratory failure needing ECMO. 

3) Was BAS done before ECMO was considered ? Were the patients reassessed after BAS for clinical improvement ? Did any patient who did not require ECMO, underwent BAS ?

This is a good point. BAS was considered for clinical improvement to decompress the left atrium to improve the pulmonary venous congestion and edema. In retrospect it would have been good to have tried this prior to ECMO but none of the patients had BAS done prior to ECMO. The primary reason was the rapid clinical deterioration in this cohort. It was hard to identify who would deteriorate and who would get better with IVIG etc. ( added reference for the entire cohort - reference 6,7) I have also added this to the discussion.

4) What was the response of Steroids/ IVIG among all patients ?

Most MISC patients responded to IVIG - there was a slight bump in BNP and troponins in some patients possibly due to the volume overload from IVIG but they recovered. 

5) Was Kawasaki's disease considered in any patients ?

Yes. This was a diagnostic dilemma in a few patients in the entire cohort and was considered in patient 1 but based on the overall picture and markers was considered MISC.

6) Did any patient who had significant LV dysfunction at presentation  were treated without ECMO?

Yes there were a few patients with an EF as low as 35% that recovered with IVIG and did not need ECMO.